# Low Antibody-Dependent Enhancement of Viral Entry Activity Supports the Safety of Inactivated SARS-CoV-2 Vaccines

**DOI:** 10.3390/vaccines13040425

**Published:** 2025-04-18

**Authors:** Xiaofang Peng, Yuru Han, Song Xue, Yunjiao Zhou, Weiyu Jiang, Anqi Xia, Wei Wu, Yidan Gao, Fan Wu, Qiao Wang

**Affiliations:** 1Key Laboratory of Medical Molecular Virology (MOE/NHC/CAMS), Shanghai Institute of Infectious Disease and Biosecurity, Shanghai Frontiers Science Center of Pathogenic Microorganisms and Infection, School of Basic Medical Sciences, Fudan University, Shanghai 200040, China; 22211020016@m.fudan.edu.cn (X.P.); hyr1968274346@163.com (Y.H.); 17301010038@fudan.edu.cn (S.X.); 21111010061@m.fudan.edu.cn (W.J.); 21111010079@m.fudan.edu.cn (A.X.); 13061915085@163.com (W.W.); 20211010051@fudan.edu.cn (Y.G.); 2Fundamental Research Center, Shanghai Yangzhi Rehabilitation Hospital (Shanghai Sunshine Rehabilitation Center), School of Medicine, Tongji University, Shanghai 201619, China; centrallab5th@163.com

**Keywords:** SARS-CoV-2, monoclonal antibody, antibody-dependent enhancement, epitopes, inactivated vaccination, convalescence

## Abstract

Background/Objectives: The antibody-dependent enhancement (ADE) of viral entry has been documented for SARS-CoV-2 infection both in vitro and in vivo. However, the potential for the SARS-CoV-2 vaccination to elicit similar ADE effects remains unclear. Methods: In this study, we assessed the in vitro ADE potential of monoclonal antibodies (mAbs) derived from individuals vaccinated with the inactivated SARS-CoV-2 vaccine and compared them to those from one convalescent donor. Results: Our analysis revealed no significant difference in binding affinity or neutralizing capacity between the vaccinated and convalescent mAbs. However, the inactivated SARS-CoV-2 vaccination induced fewer ADE-inducing mAbs, particularly those targeting the Class III epitope on the receptor-binding domain (RBD) compared to those from the convalescent individual. Moreover, no significant in vitro ADE was detected in either vaccinated or convalescent sera, indicating low levels of ADE-inducing antibodies in the sera. Conclusions: An inactivated SARS-CoV-2 vaccination induces fewer ADE-inducing antibodies compared to natural infection, further emphasizing the safety of inactivated SARS-CoV-2 vaccines.

## 1. Introduction

Antibody-dependent enhancement (ADE) is a phenomenon where pre-existing antibodies enhance viral infection through the interaction of virus–antibody immune complexes with Fc/complement receptors [1,2]. This antibody-dependent receptor-mediated endocytosis facilitates “ADE of viral entry” into the host cells. Subsequent viral replication and the secretion of proinflammatory factors may promote pathological damage, leading to a series of negative consequences known as “ADE of disease” [3].

ADE has been identified in many types of viruses, including dengue [4], Zika [5,6], human immunodeficiency virus (HIV) [7], and SARS-CoV [8,9,10]. During the COVID-19 pandemic, numerous studies, including ours, have confirmed the ADE of viral entry in vitro for SARS-CoV-2 in an Fc gamma receptor (FcγR)-dependent manner [11,12] using Raji cell- or monocyte-based ADE assays [8,13,14]. These studies raised concerns about the safety of vaccination and antibody therapeutics against SARS-CoV-2. However, clinical observations and macaque experiments showed that antibodies with ADE activity for viral entry in vitro did not necessarily cause “ADE of disease” in vivo [15,16], though the underlying mechanism remains elusive.

During the Coronavirus disease 2019 (COVID-19) pandemic, the worldwide effort to develop multiple SARS-CoV-2 vaccines, including inactivated vaccines, mRNA vaccines, recombinant protein vaccines, and adenovirus-based vaccines, has significantly reduced SARS-CoV-2 infection, severe disease, and mortality rates in the naïve population and contributed tremendously to controlling the pandemic.

While several studies have explored ADE in the context of natural SARS-CoV-2 infection [13,14,17,18,19,20], the impact of vaccination, particularly with inactivated vaccines, on ADE remains underexplored. Here, this study aims to fill this gap by understanding the vaccination-induced ADE and evaluating the potential ADE effect of inactivated COVID-19 vaccines. We assessed the ADE potential of viral entry in vitro for monoclonal antibodies (mAbs) derived from inactivated vaccinated individuals and compared them with those from a SARS-CoV-2 convalescent donor. By exploring the factors influencing ADE, we hope to provide insights into the differences in the antibody immune response elicited by inactivated SARS-CoV-2 vaccination and SARS-CoV-2 natural infection and better address the concern of potential ADE risk for SARS-CoV-2 vaccination.

## 2. Materials and Methods

### 2.1. Human Sample and Blood Collection

The participants in this study included two types of volunteers: 49 individuals who had received an inactivated SARS-CoV-2 vaccination without any SARS-CoV-2 infection, and 28 convalescent individuals who had recovered from SARS-CoV-2 infection. Among these donors, four vaccinated individuals [21] and one convalescent individual [12] provided consent for blood donation aimed at single-B cell-based antibody cloning. At the time of blood donation, the four vaccinated individuals had not been infected by SARS-CoV-2, while the one convalescent donor had not been vaccinated at all. Volunteer recruitment and blood collection were conducted at Fudan University. All procedures involving human participants were approved by the Fudan University School of Basic Medical Sciences Ethics Committee (approval number: 2022-C005), and all participants provided informed consent. Serum samples collected were heat-inactivated at 56 °C for 30 min, aliquoted, and stored at −80 °C for subsequent analyses.

### 2.2. Antibody Preparation

Monoclonal antibodies derived from the four inactivated vaccinated individuals [21] and one convalescent individual [12] have been previously detailed. These antibodies were expressed in mammalian HEK293F cells, following established protocols, and purified using protein G resin (GenScript, Nanjing, China) according to the manufacturer’s guidelines.

### 2.3. Generation of SARS-CoV-2 Pseudotyped Viruses

SARS-CoV-2 pseudotyped viruses were generated by well-established methodologies as previously described. Briefly, the pNL4-3.Luc.RE backbone plasmid and the pcDNA3.1-SARS-CoV-2-S plasmid, encoding the S protein, were co-transfected into HEK293T cells using a Vigofect transfection reagent (Vigorous Biotechnology, Beijing, China). After a four-hour incubation, the medium was replaced with a fresh Dulbecco’s Modified Eagle’s Medium (DMEM, Thermo Fisher Scientific, Waltham, MA, USA) supplemented with a 10% heat-inactivated fetal bovine serum (FBS, Thermo Fisher Scientific, Waltham, MA, USA), and the cells were incubated for an additional 48 h. The conditioned medium was then collected, centrifuged to remove cellular debris, filtered through a 0.22 µm membrane (Millipore, Darmstadt, Germany), and stored at −80 °C for subsequent in vitro neutralization and ADE assays.

### 2.4. In Vitro Antibody-Dependent Enhancement (ADE) Assays

The in vitro ADE assay was performed as previously reported [12]. Raji cells, which express FcγRII and are derived from a Burkitt’s lymphoma patient, were utilized for the in vitro ADE assay [8,12]. Raji cells were seeded in 96-well plates coated with 0.1% poly-L-lysine (Solarbio, Beijing, China) in Phosphate-buffered saline (PBS, Solarbio, Beijing, China) and cultured for 24 h at 37 °C. The pseudovirus concentration was standardized to 2 × 10^4^ copies/µL based on quantitative real-time PCR. Antibodies were three-fold serially diluted (nine dilutions in total) from a maximum concentration of 100 µg/mL, mixed with viral soup, and incubated for 30 min at 37 °C. The mixture was then added to the cultured Raji cells and incubated for 60 h. Luciferase activity was measured using the Firefly Luciferase Assay Kit (Promega, Madison, WI, USA) and normalized against an ADE control antibody, XG043 (2 µg/mL) [12]. In other words, the relative ADE activity of 2 µg/mL XG043 was set as 1 and used as a cutoff for the ADE curves. The average normalized luciferase activity, calculated from three replicates for each antibody, indicated the level of in vitro ADE of viral entry.

### 2.5. In Vitro Neutralization Assay

In vitro neuralization assays were conducted using pseudotyped viruses as previously reported [12]. Briefly, Huh-7 cells were seeded in 96-well plates and cultured for 8 h at 37 °C. Antibodies were three-fold serially diluted (nine dilutions in total) from a maximum concentration of 10 µg/mL. Antibodies were then mixed with pseudoviral soup and incubated for 30 min at 37 °C prior to infection of culture Huh-7 cells. After 12 h, the medium containing antibodies and pseudoviruses was replaced with fresh DMEM containing 10% FBS. An additional 24 h later, cells were harvested, and luciferase activity was measured using a Firefly Luciferase Assay Kit and luminometer (VICTOR Nivo, Revvity, Bethesda, MD, USA) following the manufacturer’s instructions.

### 2.6. ELISA

To assess antibody binding affinity, ELISA was conducted as previously reported [12]. Briefly, 96-well plates were coated with 10 µg/mL SARS-CoV-2 S protein ectodomain (S-ECD) and incubated overnight at 4 °C. Tested antibodies were added at a three-fold serial dilution (eight dilutions in total), starting from a maximum concentration of 10 μg/mL, and incubated for one hour at room temperature. For signal detection, an HRP-conjugated goat anti-human IgG antibody (Thermo Fisher Scientific, Waltham, MA, USA) was added and incubated for one hour. The antigen binding capacity was finally determined by calculating the area under the curve (AUC) value for each antibody using Graphpad PRISM software (Version 9.5.0), based on optical density (OD) readings.

### 2.7. Competition ELISA

To map the antibody epitopes on the SARS-CoV-2 S protein, competition ELISAs were executed using monoclonal antibodies as previously described [12]. Briefly, SARS-CoV-2 S protein ectodomain (S-ECD) at a concentration of 2 µg/mL was applied to coat 96-well plates. Unbiotinylated RBD- or NTD-binding antibodies (15 µg/mL), designated as the 1st antibody, were utilized for blocking at room temperature for two hours. Subsequently, the 2nd biotinylated detection antibody (0.25 µg/mL) was introduced into the wells and incubated for an additional 30 min at room temperature. Detection of the 2nd biotinylated antibody binding was facilitated by streptavidin-HRP, incubated for one hour at room temperature. PBS served as a control for normalization by replacing the 1st blocking antibody. For the classification of RBD epitopes among XGv antibodies derived from vaccinated donors, four XG antibodies, cloned from a convalescent donor [12], were employed as the 2nd detection antibody: XG017 for Class I, XG041 for Class II, XG014 for Class III, and XG011 for Class IV. These antibodies’ non-overlapping RBD epitope groups were previously characterized and designated as Groups I to IV, [12] aligning with RBD Classes IV, II, I, and III, respectively, as reported [22].

The serum competition ELISA was conducted analogously. The 96-well plate was coated with 1 µg/mL SARS-CoV-2 RBD recombinant protein and incubated overnight at 4 °C. Serum samples, diluted 1:4 in PBS, were then added and incubated for 2 h at room temperature. The 2nd biotinylated XG004 antibody (RBD Class III), at a final concentration of 0.2 µg/mL, was then directly added for a 30 min incubation at room temperature. Streptavidin-HRP was used for detection. PBS was employed as a normalization control by substituting serum samples. Additionally, another RBD Class III antibody, XG003, served as a control to assess the blocking efficiency against the 2nd biotinylated XG004 antibody, contrasting with XG008, an RBD Class I antibody, which did not exhibit blocking activity against XG004.

### 2.8. Statistical Analysis

The details of statistical analysis are shown within the results section and figure legends. All statistical computations were executed utilizing GraphPad PRISM software (Version 9.5.0). The threshold for statistical significance was set at *p* < 0.05.

## 3. Results

### 3.1. Antibodies from Vaccinated Volunteers

Our previous study, along with a wealth of data from other researchers, has documented the presence of the antibody-dependent enhancement (ADE) of viral entry for SARS-CoV-2 in both in vitro and in vivo settings [11,12,16,20]. In these studies, however, the antibodies or sera that induced ADE were derived from individuals who had been infected or recovered, thus highlighting the impact of infection-induced antibodies. However, it remains unclear whether vaccination-induced antibodies could elicit similar ADE effects.

To investigate the vaccination-induced ADE effect, in this study, we aimed to assess the in vitro ADE potential of monoclonal antibodies (mAbs) derived from four vaccinated individuals, who had received the inactivated SARS-CoV-2 vaccine and remained uninfected, and to compare their ADE activity with mAbs derived from a convalescent donor who had recovered from SARS-CoV-2 infection.

From four vaccinated individuals, we cloned in total 422 antibody variable region genes with naturally paired heavy and light chains [21]. Many closely related antibody clones with the same immunoglobulin variable genes and highly similar CDR3 sequences were identified (Figure 1A). From the closely related antibody clones, we selected 50 representative mAbs, designated as XGv001–XGv050 (Figure 1A), and expressed these antibodies in mammalian HEK293F cells (see Section 2). Only XGv037 and XGv048 with extremely low yield were excluded in the following studies.

To compare with antibodies isolated from the convalescent donor, we used 45 mAbs isolated from a SARS-CoV-2 wildtype-infected but recovered individual [12] (Figure 1B). These convalescent antibodies were designated as XG001–XG048, while XG021, XG034, and XG039 were excluded due to their lack of binding capacity against S-ECD [12] (Figure 1B).

### 3.2. Distinct ADE Levels Induced by Convalescence- and Vaccination-Derived Antibodies

We assessed the ADE effects of these mAbs using luciferase-expressing wildtype SARS-CoV-2 pseudoviruses to infect Raji cells (Appendix A), which express FcγRII and are derived from a Burkitt’s lymphoma patient and were previously utilized for the in vitro ADE assay for SARS-CoV-1 [8,12]. The antibody-induced viral entry level, measured by the activity of the ectopically expressed luciferase reporter in Raji cells, varied significantly, with some mAbs showing strong enhancement while others showed none (Figure 2A,B).

Notably, none of the 48 vaccination-induced XGv mAbs showed relative ADE values above the cutoff, defined as one by the relative ADE activity of an ADE control antibody, XG043 (2 µg/mL) (Figure 2A). However, on the other hand, for convalescence-induced mAbs, 9 out of 45 XG mAbs promoted SARS-CoV-2 pseudovirus infection, above the cutoff, in Raji cells (Figure 2B). In particular, among the nine ADE-positive XG antibodies, three mAbs, XG005, XG016, and XG003, were able to induce ADE across the entire concentration range tested (0.015–100 µg/mL) (Figure 2B).

To better compare the ADE activity, we calculated the maximum relative luciferase activity (enhancing power) and the area under the ADE curve (ADE AUC) for each tested antibody (Appendix A). The enhancing power and ADE AUC values were highly correlated, with correlation coefficients of 0.8940 (*p* < 0.0001) for vaccination-induced XGv mAbs and 0.8809 (*p* < 0.0001) for infection-induced XG mAbs (Appendix A).

Regarding enhancing power, while none of the vaccination-induced XGv mAbs exhibited significant ADE, 22% (10 out of 45) of the XG mAbs from the convalescent donor induced strong ADE of viral entry, with enhancing power exceeding the cutoff value of one, as set by the ADE control antibody, XG043 (2 µg/mL) (Figure 2C,D). In terms of ADE AUC, only one mAb from vaccinated individuals, XGv017, showed an ADE AUC value greater than 40, compared to 9 out of 45 XG mAbs from the convalescent donor (Figure 2C,D). Statistically, vaccination-induced XGv mAbs exhibited significantly lower ADE enhancing power (*p* = 0.0216, Figure 2G) and ADE AUC values (*p* = 0.0421, Figure 2H) compared to infection-induced mAbs.

Collectively, these findings indicate that antibodies induced by inactivated SARS-CoV-2 vaccination resulted in a lower level of ADE of viral entry compared to antibodies induced by SARS-CoV-2 infection.

Since XG mAbs were isolated from only one convalescent donor, to further strengthen our conclusion, we then randomly selected 15 previously published antibodies isolated from 14 convalescent individuals (Appendix A) [[22],[23],[24],[25],[26],[27],,[28],[29],[30],[31],[32]] and performed in vitro ADE assays (Appendix A). The ADE results showed that 6 out of the tested 15 antibodies (40%) induced strong ADE of viral entry with enhancing power exceeding the cutoff value of one (Appendix A) and that 5 out of the tested 15 antibodies showed ADE AUC values greater than 40 (Appendix A). Similar to XG mAbs, the selected 15 mAbs exhibited significantly higher ADE enhancing power (*p* = 0.0001, Appendix A) and ADE AUC values (*p* < 0.0001, Appendix A) compared to the vaccination-induced XGv mAbs.

### 3.3. Lack of Correlation Between Neutralization/Binding Affinity and ADE Effect

Having established the differential ADE potential of mAbs derived from vaccinated and convalescent individuals, we next sought to investigate whether these differences could be attributed to variations in the neutralization potency or binding affinity of these antibodies.

We analyzed the binding capacity of antibodies against the SARS-CoV-2 wildtype S protein for both XGv (Figure 3A,B) and XG antibodies (Figure 3C,D). A statistical comparison revealed no significant difference in binding levels between convalescent and vaccinated antibodies (Figure 3E). Correlation analysis between ADE enhancing power and ELISA–AUC values showed no significant correlation, with correlation coefficients of 0.1184 (*p* = 0.2582) (Figure 3F), suggesting that antibody-mediated enhancement of viral entry is not directly related to antibody binding.

Similarly, we compared the neutralization potency, as measured by IC_50_ values from in vitro neutralization assays (Figure 3G), and found no significant difference between the two types of antibodies (*p* = 0.7756) (Figure 3H). Additionally, there was no significant association between ADE enhancing power and neutralization potency (r = 0.1447, *p* = 0.1869) (Figure 3I).

### 3.4. Epitope Association for ADE Antibodies

Many epitopes on the SARS-CoV-2 S protein, particularly within the receptor-binding domain (RBD), have been identified and characterized [22,33,34]. Given our previous findings that ADE antibodies are associated with specific RBD epitopes [12], we sought to investigate whether the differences in ADE activity between infection- and vaccination-induced antibodies could be attributed to distinct epitopes recognized by the XG and XGv antibodies.

To characterize the epitope for each XG and XGv antibody, we utilized competition ELISA. Since the epitopes of XG antibodies have been previously characterized [12], we focused on determining the epitopes for XGv antibodies. The competition ELISA results identified four mutually exclusive RBD epitopes (RBD Class I, II, III, and IV as previously reported [12,22]) for 26 RBD–binding XGv antibodies [35] (Appendix A). Similarly, competition ELISA using N-terminal domain (NTD)–binding XGv antibodies revealed three non-overlapping epitopes on NTD (NTD Group I, II, and III) (Appendix A).

Next, we compared XG and XGv antibodies that recognize the same RBD epitopes and observed a significant ADE effect, particularly among RBD Class III XG antibodies (7 out of 23 RBD–binding XG antibodies) (Figure 4A). Notably, 5 out of 7 (71%) RBD Class III XG antibodies were ADE-positive. In contrast, among the 26 RBD–binding XGv mAbs isolated from vaccinated individuals, only two mAbs, XGv016 and XGv042, recognized the RBD Class III epitope. Importantly, these two RBD Class III XGv antibodies induced significantly lower levels of in vitro ADE of viral entry (*p* = 0.0278) compared to the seven RBD Class III XG antibodies (Figure 4B). Moreover, XG and XGv antibodies recognizing other RBD epitopes, three non-overlapping NTD epitopes, and the non-RBD/NTD epitope showed no statistical differences in their ADE potential (Figure 4B–D).

Together, these results suggested that SARS-CoV-2 infection generates more RBD Class III mAbs with higher ADE potential compared to inactivated SARS-CoV-2 vaccination. These findings explain the lower prevalence of ADE antibodies in inactivated vaccinated individuals relative to those who have recovered from SARS-CoV-2 infection.

### 3.5. Lower Levels of RBD Class III Antibody in Vaccinated Sera

Given that our data demonstrated fewer ADE-positive RBD Class III mAbs induced by inactivated vaccines compared to SARS-CoV-2 infection, we then would like to compare the serological level of RBD Class III antibodies in both vaccinated and convalescent sera. To quantify these antibodies in sera, we conducted competition ELISAs using sera from 45 vaccinated and 27 convalescent individuals, along with a second biotinylated RBD Class III mAb, XG004 [12], for detection (Figure 5A). Two scenarios were considered as follows: (1) a high abundance of RBD-binding Class III mAbs in sera would effectively block the second biotinylated RBD Class III mAb, XG004, and (2) a lack of serological RBD–binding Class III mAbs would result in a strong signal from the second biotinylated RBD Class III mAb, XG004 (Figure 5A).

As anticipated, when used as the first blocking antibody, XG003, another Class III mAb recognizing the RBD, strongly inhibited the binding of the second biotinylated RBD Class III mAb, XG004, to the coated SARS-CoV-2 RBD protein (Figure 5B). In contrast, both PBS and XG008, an RBD Class I mAb, failed to inhibit this binding (Figure 5B).

From the serological competition ELISAs, we calculated the normalized OD values for both vaccinated and convalescent sera (Figure 5B). The signal generated by the second biotinylated RBD Class III mAb, XG004, was maintained at 30–99% in vaccinated sera, compared to only 6–45% in convalescent sera. Statistical analysis revealed significantly higher XG004–inducing signals in vaccinated sera compared to convalescent sera (*p* < 0.0001) (Figure 5C),

Together, these results suggested that, compared to inactivated vaccinated sera, convalescent sera contain higher levels of RBD Class III antibodies, which are associated with a greater potential for ADE of viral entry in vitro.

### 3.6. Comparable Serological ADE Activity Between Vaccination and Convalescence

Given that convalescent sera contain higher levels of ADE-positive RBD Class III antibodies, we hypothesized that, compared to inactivated vaccinated sera, convalescent sera might induce stronger ADE of viral entry in vitro.

To test this hypothesis, we conducted serological ADE assays using serum samples in place of mAbs (Figure 6A). Based on the luciferase reading results, only weak ADE activity was observed for a few serum samples at specific concentrations (Figure 6A). Importantly, both vaccinated and convalescent sera exhibited relative ADE values below the cutoff (defined as one by the 2 µg/mL ADE-positive XG043 reference antibody [12]) across all tested serum concentrations (Figure 6B). Moreover, no significant difference was observed in ADE AUC and ADE enhancing power between vaccinated sera and convalescent sera (Figure 6C,D).

These serological ADE data contradict our initial hypothesis that convalescent sera would induce stronger ADE of viral entry in vitro than vaccinated sera. Since ADE-positive mAbs tend to induce ADE of viral entry at antibody concentrations higher than 0.1 µg/mL (Figure 2B), this discrepancy may be attributed to the low concentrations of serological ADE mAbs, which are insufficient to elicit strong serological ADE.

In conclusion, our results suggest that SARS-CoV-2 vaccination induces fewer ADE mAbs targeting RBD Class III. Although more ADE-positive RBD Class III mAbs were cloned from convalescent individuals and higher levels of such mAbs were present in convalescent sera, the concentration of ADE-positive mAbs in both inactivated vaccinated sera and convalescent sera is too low to elicit strong serological ADE. This finding further emphasized the safety of inactivated SARS-CoV-2 vaccines.

## 4. Discussion

The findings of our study provide valuable insights into the ADE phenomenon in the context of SARS-CoV-2 infection and vaccination. The results of our in vitro ADE assays revealed that while there was no significant difference in binding affinity or neutralizing capacity between the mAbs derived from individuals vaccinated with the inactivated SARS-CoV-2 vaccine and those from a convalescent donor, the vaccinated individuals produced fewer ADE-inducing mAbs, particularly those targeting the RBD Class III epitope on the S protein.

This observation is significant as it suggests that the inactivated SARS-CoV-2 vaccine may elicit a different immune response compared to natural infection, potentially reducing the risk of ADE. The lower prevalence of ADE-inducing antibodies in vaccinated individuals could be attributed to the composition of the inactivated vaccine and the way it stimulates the immune system. Specifically, all four donors for antibody isolation received the SARS-CoV-2 inactivated vaccine, CoronaVac, which contains β-propiolactone/formaldehyde-inactivated SARS-CoV-2 antigens (CZ02 strain) [36,37,38]. The inactivation procedure might alter the molecular structure of viral proteins, affecting the immunogenicity of certain epitopes. Therefore, unlike natural infection, which exposes the immune system to the entire virus, the inactivated vaccine may present a more limited set of antigens, leading to a shifted immune response that avoids the production of antibodies targeting ADE epitopes, such as RBD Class III. Moreover, whether mRNA vaccines share similar ADE patterns with the inactivated vaccine needs further investigation.

Our data indicate that an antibody’s epitope specificity, rather than its ability to neutralize or bind to the virus, correlates with its ADE potential, underscoring the complexity of the ADE mechanism. Specifically, a higher prevalence of RBD Class III mAbs in convalescent individuals is associated with a higher level of ADE potential, compared to those vaccinated with the inactivated SARS-CoV-2 vaccine. These observations raise further fundamental questions about the underlying molecular mechanism related to this epitope-ADE correlation. Yet, little is known.

During antibody-mediated immune enhancement, the FcγR, which is activated by the engagement of the Fc domains on antibodies, initiates signaling to upregulate pro-inflammatory cytokines and downregulate anti-inflammatory cytokines [3]. Recently, RNA-sequencing (RNA-seq) experiments and a Gene Ontology analysis showed that strong ADE of SARS-CoV-2 upregulated the “interferon alpha/beta signaling pathway”, “negative regulation of viral genome replication”, “response to interferon-alpha pathway”, and “type II interferon signaling pathway”, causing excessive activation of the immune cascade [14]. Based on these findings, it would be essential to study in the future how RBD Class III antibodies, but not other non-ADE-inducing antibodies, mediate the downstream immunological signaling pathways.

Furthermore, our study found that neither vaccinated nor convalescent sera induced significant in vitro ADE of viral entry. This finding is particularly important as it suggests that despite the presence of ADE-inducing mAbs in convalescent sera, the concentration of such mAbs is low, resulting in an overall low level of serological ADE activity. This finding further alleviates concerns about the ADE risk associated with SARS-CoV-2 vaccination and enhances the safety profile of inactivated SARS-CoV-2 vaccines.

In the context of antibody therapy, these findings highlight the importance of considering ADE [39]. High concentrations of antibodies used in therapeutic settings could potentially enhance viral entry into cells expressing FcγRs, such as monocytes and macrophages, leading to increased inflammation and disease severity. To mitigate this risk, many anti-SARS-CoV-2 therapeutic antibodies have undergone Fc engineering, such as LALA [40,41], GRLR [42], TM [43], and others, to reduce Fc receptor-binding affinity, thereby eliminating the negative effects of ADE.

The limitation of this study is that we only recruited one convalescent donor and all the XG antibodies were from this one donor. We recruited more convalescent donors later, but they had been vaccinated. Therefore, in this study, to compare the antibodies induced by natural infection and vaccination, we decided to only use XG antibodies from the convalescent donor without receiving an inactivated vaccine.

In this study, we primarily assessed the ADE potential of viral entry. However, ADE is not only limited to enhanced viral entry but also might exacerbate disease severity through other mechanisms, such as antibody-dependent serum resistance, enhancement of inflammation, and anti-cytokine autoantibodies [44]. Antibodies can also increase disease severity during infection by enhancing pathological inflammation. For instance, afucosylated IgG was correlated with increased IL-6 and C-reactive protein levels in plasma, severe disease, and hospitalization [45,46,47].

## 5. Conclusions

Our results suggest that an inactivated SARS-CoV-2 vaccination induces fewer ADE-inducing mAbs compared to natural infection, and the overall serological ADE activity is low. These findings have important implications for the safety of COVID-19 inactivated vaccines and for alleviating public health concerns regarding SARS-CoV-2 vaccination strategies. Future studies should continue to explore the immune responses elicited by different types of SARS-CoV-2 vaccines and monitor for any potential ADE effects in vaccinated and/or infected populations.

## Figures and Tables

**Figure 1 vaccines-13-00425-f001:**
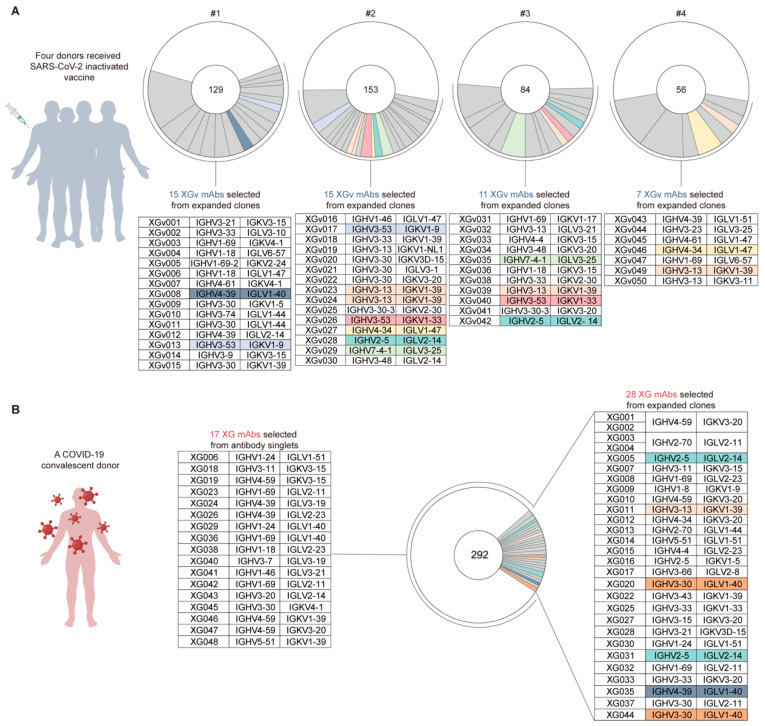
Antibodies from inactivated SARS-CoV-2 vaccinated and convalescent individuals. (**A**,**B**) A total of 48 XGv and 45 XG antibodies from four individuals (designated as #1, #2, #3 and #4) who received three doses of inactivated SARS-CoV-2 vaccine and one convalescent individual, respectively. Each pie chart represents a donor, and the number of amplified antibodies is indicated at the center of each pie chart. Within the pie chart, each gray or colored slice represents the antibodies with the same IGHV/IGLV combination and closely related CDR3s, while the big white slice represents the antibody singlets. The size of each slice is proportional to the corresponding antibody number. Similar antibody clones with the same IGHV/IGLV combination are colored among distinct donors.

**Figure 2 vaccines-13-00425-f002:**
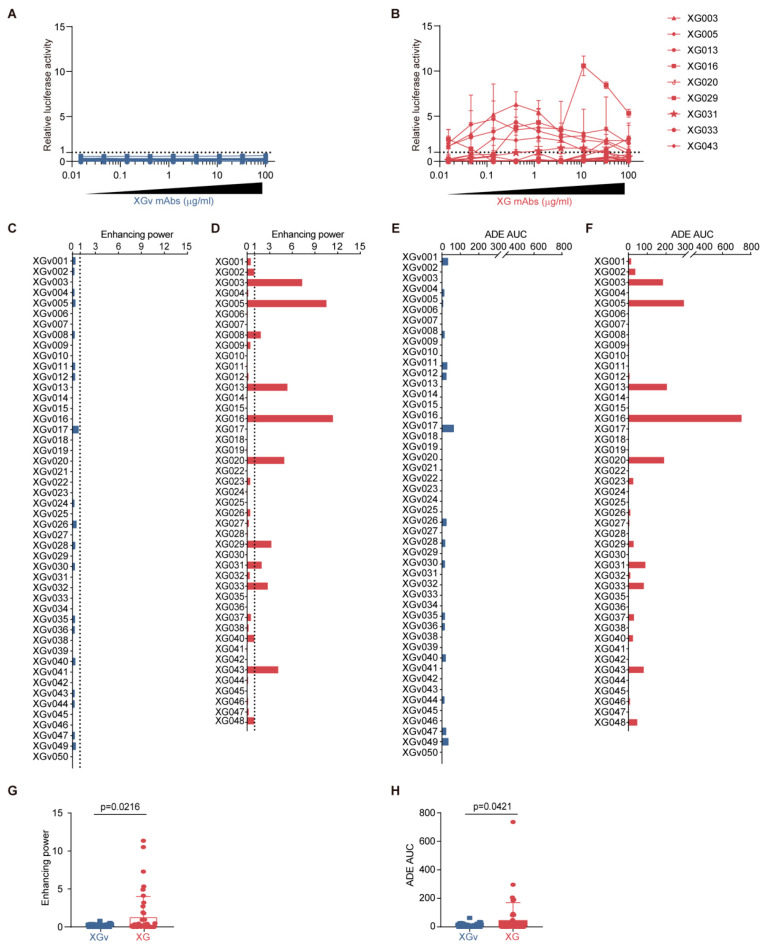
In vitro ADE activity of vaccinated and convalescent antibodies. (**A**,**B**) In vitro ADE assays. Relative luciferase activity (*y*-axis) indicates the SARS-CoV-2 pseudoviral entry level induced by various concentrations (*x*-axis) of 48 XGv vaccinated mAbs (**A**) and 45 XG convalescent mAbs (**B**). An ADE antibody, XG043 (2 µg/mL), was used as a reference for normalization, and its luciferase reading value was set as 1 and used as a cutoff for the ADE curves (dotted line). (**C**,**D**) ADE enhancing power for XGv vaccinated mAbs (**C**) and XG convalescent mAbs (**D**). (**E**,**F**) ADE AUC values: XGv vaccinated mAbs (**E**) and XG convalescent mAbs (**F**). (**G**,**H**) Statistical comparison of enhancing power (**G**) and ADE AUC (**H**) between XGv and XG mAbs. The *p* values were calculated using the Mann–Whitney test.

**Figure 3 vaccines-13-00425-f003:**
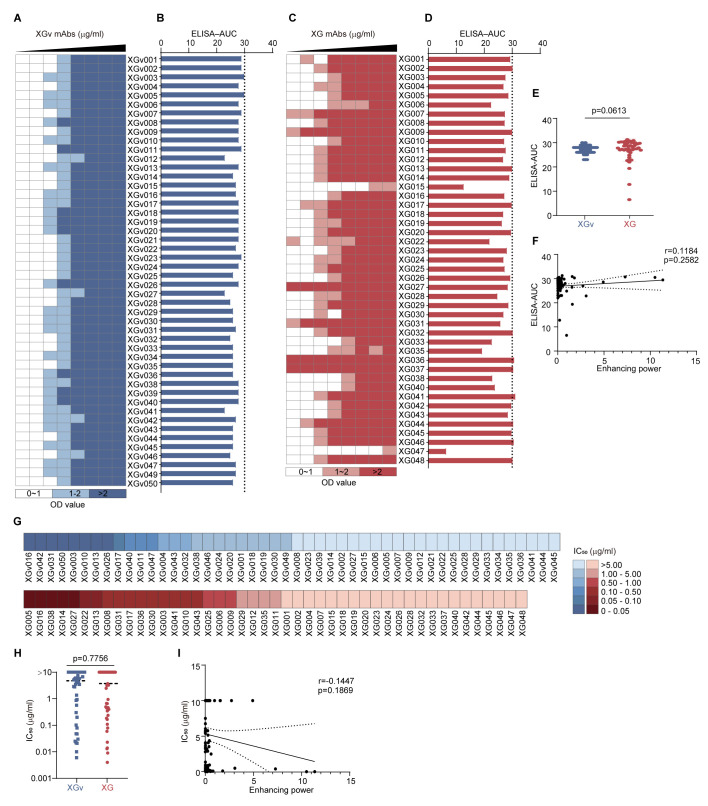
ADE activity showed no significant correlation with ELISA binding capacity and neutralization potency. (**A**,**C**) ELISA binding capacity of XGv (**A**) and XG (**C**) antibodies. Each row represents an antibody. Each antibody was 3-fold serially diluted with 8 dilutions in total (each column), with the maximum antibody concentration being 10 µg/mL. The colors of each cell in the heatmap represent the range of measured ELISA OD values: white, 0–1; light blue (**A**) and light red (**B**), 1–2; blue (**A**) and red (**B**), >2. (**B**,**D**) ELISA area under the curve (AUC) of XGv (**B**) and XG (**D**) antibodies. (**E**) Statistical comparison of ELISA–AUC between XGv and XG mAbs. The *p* value was calculated using the Mann−Whitney test. (**F**) Correlation between ADE enhancing power (*x*-axis) and ELISA–AUC (*y*-axis). Each dot represents one mAb. A linear regression model was used for correlation analysis. The 95% confidence interval of the regression line is shown by the dotted line. (**G**) IC_50_ values of XGv and XG mAbs determined by in vitro neutralization assays against wildtype SARS-CoV-2 pseudovirus. (**H**) Statistical comparison of neutralization potency (IC_50_ value) between XGv and XG mAbs. Each dot represents an antibody. The *p* value was calculated using the Mann−Whitney test. (**I**) Correlation between ADE enhancing power (*x*-axis) and IC_50_ values against wildtype SARS-CoV-2 pseudovirus (*y*-axis). Each dot represents one mAb. A linear regression model was used for correlation analysis. The 95% confidence interval of the regression line is shown by the dotted line.

**Figure 4 vaccines-13-00425-f004:**
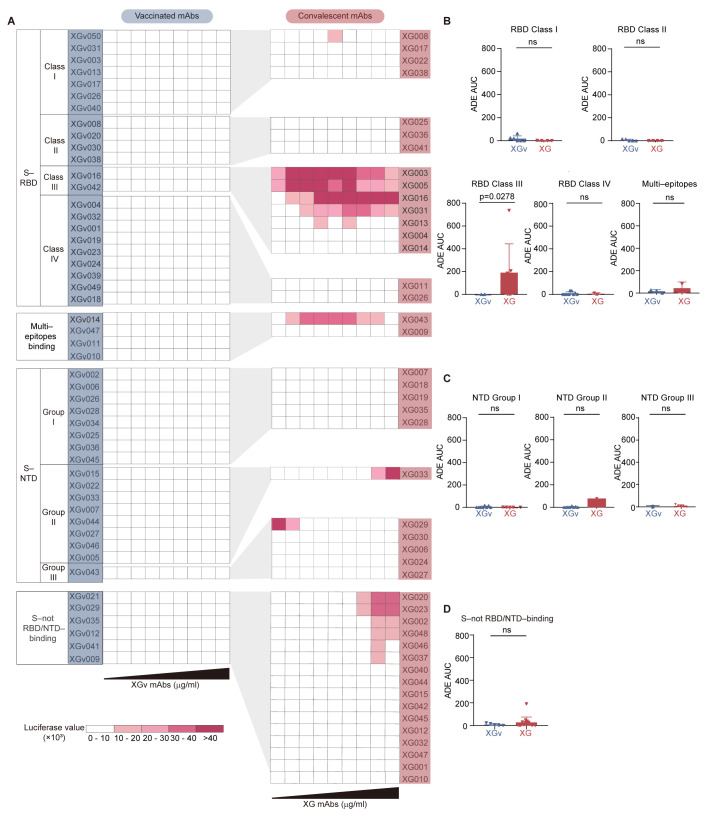
Comparison of antibody epitope and the corresponding ADE potential between XGv vaccinated and XG convalescent antibodies. (**A**) Comparison of in vitro ADE levels between XG and XGv mAbs sharing similar epitopes. The RBD–targeting mAbs include Class I, II, III, and IV antibodies and antibodies targeting cross-Class epitopes (Appendix A). The NTD–targeting mAbs include Group I, II, and III antibodies (Appendix A). There are also S–binding antibodies not targeting RBD and NTD. (**B**) Statistical comparison of ADE AUC between RBD–binding XGv and XG mAbs. (**C**) Statistical comparison of ADE AUC between NTD–binding XGv and XG mAbs. (**D**) Statistical comparison of ADE AUC between non-RBD/NTD–binding XGv and XG mAbs. Statistical analysis was performed using the Mann–Whitney test. n.s., not significant, *p* > 0.05.

**Figure 5 vaccines-13-00425-f005:**
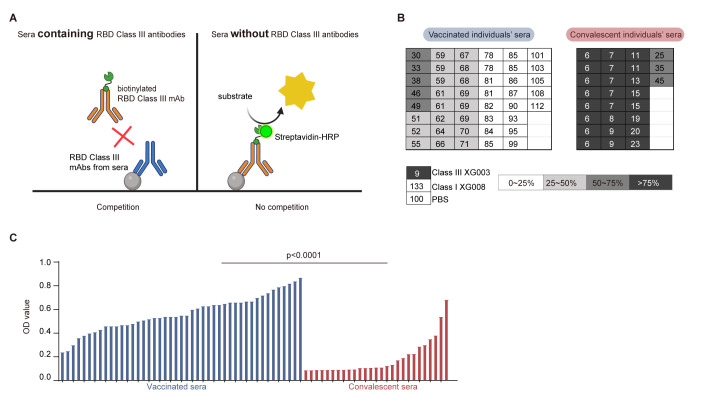
Lower levels of ADE-inducing RBD Class III mAbs in vaccinated sera. (**A**) Diagram of serological competition ELISA. (**B**) Levels of RBD Class III antibodies in vaccinated and convalescent sera measured by competition ELISAs. After blocking the coated RBD proteins using serum samples, the 2nd biotinylated RBD Class III mAb, XG004, was used for detection by streptavidin-horseradish peroxidase (HRP). XG003, another Class III mAb recognizing the RBD, strongly inhibited the binding of the 2nd biotinylated XG004; meanwhile, PBS and XG008, an RBD Class I mAb, failed to do so. (**C**) Statistical comparison of competition ELISA OD values between vaccinated and convalescent sera. The *p* value was calculated using the Mann–Whitney test.

**Figure 6 vaccines-13-00425-f006:**
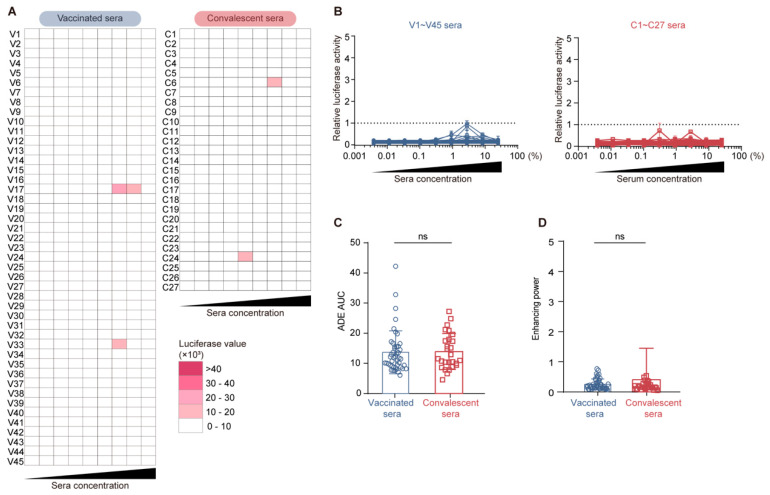
No significant serological ADE for both vaccinated and convalescent sera. (**A**) Serological ADE activity. Each line represents a vaccinated (V) or convalescent (C) donor. To measure the ADE of viral entry in vitro, 3-fold serially diluted sera (nine dilutions in total, each column) were incubated with SARS-CoV-2 pseudovirus before being administered into Raji cells. The colors of each cell in the heatmap represent the measured luciferase reading values. (**B**) Relative luciferase activity for vaccinated sera (**left**) and convalescent sera (**right**). (**C**,**D**) Statistical comparison of ADE AUC (**C**) and enhancing power (**D**) between vaccinated and convalescent sera. Each dot and block represents one serum sample from vaccinated and convalescent individuals, respectively. The *p* values were calculated using the Mann–Whitney test. ns, not significant (*p* > 0.05).

## Data Availability

The data will be available upon request.

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
