# Peer review of "Low Antibody-Dependent Enhancement of Viral Entry Activity Supports the Safety of Inactivated SARS-CoV-2 Vaccines"

_vaccines, 2025, doi:10.3390/vaccines13040425_

Round 1
Reviewer 1 Report
Comments and Suggestions for Authors
Peng et al., submitted the study entitled “Modest Prevalence and Low Serum Concentration of Antibody-Dependent Enhancement (ADE)-Positive Antibodies Underpin the Safety Profile of Inactivated SARS-CoV-2 Vaccines”.
The study investigated the potential for SARS-CoV-2 vaccines to induce antibody-dependent enhancement (ADE) effects. While the findings suggest a low prevalence of ADE-inducing antibodies in vaccinated individuals as well as convalescent patients. However, the inactivated vaccine induced moderate ADE compared to the other vaccine and convalescent strategies. The major limitation of the study is the lack of a mechanistic explanation for how inactivated vaccines might contribute to ADE. The study effectively compares ADE activity between groups but does not explore the underlying immunological pathways leading to reduced ADE risk. As such, the manuscript is acceptable for a short communication rather than a full-length article.
Reviewer 2 Report
Comments and Suggestions for Authors
The study presents some interesting results and considerations for the further development of vaccines.
While monoclonal antibody work is elegant in the end it compared the antibodies from only one individual that had recovered from a covid infection with antibodies from four subjects who underwent vaccination with an inactived vaccine. That should be emphasised in the discussion of the difference between the results with the monoclonals and sera.
Had the patients who had recovered from covid been vaccinated and was it with the inactivated protein vaccine?
Given the world-wide usage the enhancing specificities induced by mRNA vaccines would be of more interest but presumably that is another story.
Reviewer 3 Report
Comments and Suggestions for Authors
The authors examined the prevalence of antibody-dependent enhancing (ADE) antibodies in serum of SARS-CoV-2 vaccinated and convalescent individuals. They cloned and tested between 48 monoclonal antibodies isolated from 4 vaccinated individuals and 45 from a convalescent patient for ADE, neutralization, epitope specificity and binding to SARS-CoV-2 S protein. They report no ADE activity among the monoclonal antibodies isolated from the vaccinated individuals and 9 out of 45 with ADE activity for the antibodies from the convalescent patient. Evaluation of serum samples from 49 vaccinated and 27 convalescent individuals revealed minimal ADE activity and no difference between the two groups.
The authors provide data that support the conclusion that vaccination with an inactivated whole SARS-CoV-2 virus does not induce antibody-dependent enhancement of viral entry which corroborates other studies. However, the conclusion that “inactivated SARS-CoV-2 vaccination induces fewer ADE-inducing mAbs compared to natural infection” (Abstract, line 25-26) is not supported by the data as they are based on observations from just one convalescent patient. Indeed, the lack of significant ADE in serum samples from both vaccinated and convalescent individuals (Figure 6) indicates that vaccine- or infection-induced antibodies do not enhance viral entry into cells. My other main concern with the manuscript as written is that the authors do not distinguish between ADE of viral entry and ADE of disease. Antibodies can also enhance disease by other mechanisms than increased infection of cells. I refer the authors to a recent review on this subject (Wells et al., Nature Rev Immunol 2025). The in vitro assay employed in the research only measures ADE of viral entry. Thus the last sentence of the abstract (lines 26-27) “the overall serological ADE activity is low, which may explain the rarity of "ADE of disease" observed in vivo” is not correct as the serologic ADE activity only refers to ADE of viral entry. There is little evidence for ADE of viral entry with SARS-CoV-2, but some evidence of ADE of inflammation/disease during natural infections.
Other comments:
- Title: The title is a bit misleading as the authors base the prevalence on just four individuals. A suggested title would be “Low antibody-dependent enhancement of viral entry activity of serum from vaccinated individuals supports the safety of inactivated SARS-CoV-2 vaccines.”
- Abstract: “inactivated SARS-CoV-2 vaccination induces fewer ADE-inducing mAbs compared to natural infection” (Abstract, line 25-26). Replace “mAbs” by “antibodies” as vaccination does not induce monoclonal antibodies.
- Lines 226-227: “these findings indicate that antibodies induced by inactivated SARS-CoV-2 vaccination resulted in a lower level of ADE of viral entry compared to antibodies induced by SARS-CoV-2 infection.” This conclusion is not supported by the data as the results are from just one convalescent patient. Indeed, the authors should discuss this as a limitation of their study since the prevalence of ADE-mAbs from this patient may be unique to this individual.
- Line 372-373: “This observation is significant as it suggests that the inactivated SARS-CoV-2 vaccine may elicit a different immune response compared to natural infection, potentially reducing the risk of ADE.” See my previous comment (#3). The data presented in Figure 6 suggest no difference in immune response between vaccinated and convalescent individuals.
- Line 391-397. The authors should discuss the different forms of ADE. A lack of ADE of viral entry does not necessarily lead to lack of ADE of disease.
- Line 405-406. “….inactivated SARS-CoV-2 vaccination induces fewer ADE-inducing mAbs compared to natural infection”. See comment #3.
Round 2
Reviewer 1 Report
Comments and Suggestions for Authors
The authors have revised the manuscript by considering the reviewers' comments. The revised manuscript is acceptable for publication in its current form. Hereby I endorse the manuscript for publication.
Reviewer 3 Report
Comments and Suggestions for Authors
The authors have addressed my concerns and comments.